# Contrastive Class Anchor Learning for Open Set Object Recognition in Driving Scenes

**Zizhao Li**                                                     *zizhao.li1@student.unimelb.edu.au*
*The University of Melbourne*

**Kourosh Khoshelham**                                           *k.khoshelham@unimelb.edu.au*
*The University of Melbourne*

**Joseph West**                                                   *joseph.west@unimelb.edu.au*
*The University of Melbourne*

**Reviewed on OpenReview:** *https://openreview.net/forum?id=l0Uum9SJgM*

## Abstract

Conventional object recognition models operate under closed-set assumptions presuming that the training dataset is sufficiently comprehensive that any object detected during inference can be assigned to some known prior class. This assumption is flawed and potentially dangerous for real-world applications such as driving scene perception where diverse objects and unexpected behaviours should be expected. In order to progress towards trusted autonomous platforms object recognition models need Open Set Recognition (OSR) methods capable of identifying unknown classes while maintaining good performance on known classes. Existing OSR methods are mostly designed for image data and utilize generative models which are hard to train. In this paper, we propose S2CA, a Supervised Contrastive Class Anchor learning method which leverages contrastive learning principles to effectively reject unknown classes by increasing intra-class compactness and inter-class sparsity of known classes in feature space. We train a feature encoder through contrastive learning while ensuring that features of known classes form compact clusters, and then transfer the trained encoder to the OSR task. During inference, the model rejects unknown classes based on class-agnostic information in feature space and class-related information in logit space. The proposed OSR method is simple yet powerful. It is not only suitable for image-based object recognition models, but can also be used for a variety of lidar-based object recognition models. We demonstrate superior performance of S2CA when compared with state of the art methods on two widely used driving scene recognition datasets, i.e., KITTI and nuScenes. The source code is available at `https://github.com/343gltysprk/s2ca`.

## 1 Introduction

Autonomous driving has received a great deal of attention in recent years due to its potential for eliminating human error, which is the major cause of most road accidents. To ensure safe and efficient navigation, autonomous vehicles must be able to perceive their surroundings using onboard sensors and understand the driving scene. This involves localizing and recognizing surrounding objects.

Conventional object recognition models typically make the assumption of a closed set, requiring that all object categories which the model will encounter are available in the training set (Scheirer et al., 2013). When the recognition model encounters an unknown object, it will misclassify it as one of the known classes. In the real world, autonomous vehicles encounter a wide variety of objects. It is not feasible to include all object categories in the training set. For example, autonomous driving datasets designed for the northern hemisphere do not typically contain kangaroos, yet in Australia kangaroos frequently appear on the road.

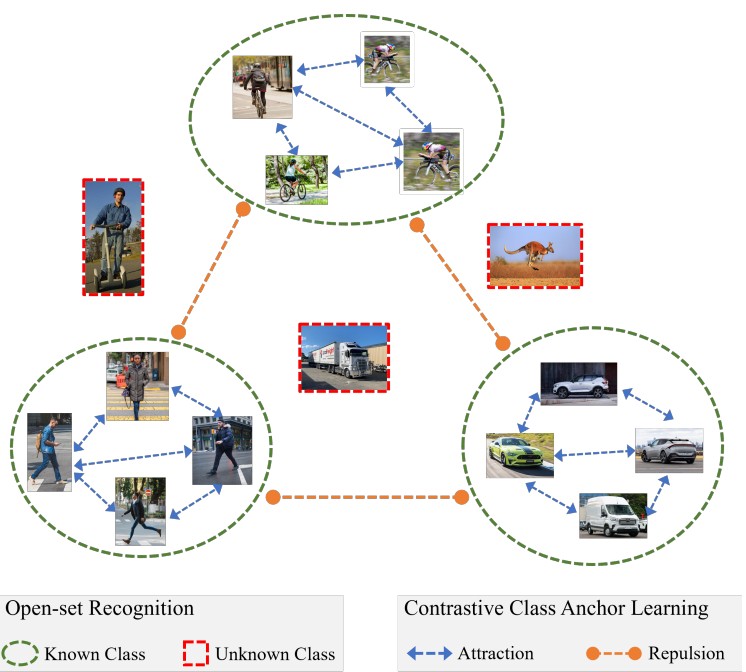

Figure 1. Our proposed approach to open-set recognition aims to effectively recognise unknown objects while correctly classifying known classes by increasing intra-class compactness and inter-class sparsity in the feature space, thereby reducing the chance of mixing known and unknown classes. This is achieved by supervised contrastive learning (SCL) which encourages intra-class attraction and inter-class repulsion. To overcome the limitation of the original SCL, which contrasts the instances only in the same mini-batch, we introduce Contrastive Class Anchor Learning, which applies a global constraint to contrastive learning by placing class anchors in the logit space. The proposed method is efficient and suitable for both image-based and lidar-based object recognition.

Therefore, the closed-set assumption will inevitably have an adverse impact on the autonomous driving system.

An ideal driving scene perception system should be able to detect all object instances on the road surface, then reject unknown objects and classify the known ones. Unknown objects can then be stored for later labelling, *e.g.* by a human supervisor. Scheirer et al. (2013) define the problem of rejecting unknown classes ($\mathcal{U}$) and simultaneously classifying known classes ($\mathcal{K}$) as open-set recognition (OSR). Unknown classes are sometimes referred to as "unknown unknowns", because they are not only unseen but also have no side information (*e.g.*, semantic attributes or known properties) during training (Oza & Patel, 2019; Geng et al., 2021).

An intuitive OSR method is to set a threshold for the classification confidence score (*e.g.* softmax probability) of the most probable class produced by the model (Hendrycks & Gimpel, 2017). However, previous research has shown that samples in $\mathcal{U}$ can also produce equally high softmax scores (Bendale & Boult, 2015). State-of-the-art OSR methods often involve generative models (Ge et al., 2017; Neal et al., 2018; Oza & Patel, 2019; Perera et al., 2020; Kong & Ramanan, 2021), which help separate known and unknown classes in the feature space. While improved feature learning leads to better OSR performance, the complexity and challenges of training generative models limit their application in open-set recognition in diverse scenarios. Miller et al. (2021) proposed Class Anchor Clustering (CAC), which forces known classes to form compact clusters around class anchors in logit space and rejects unknown objects based on their distance to known clusters. However, the cluster compactness achieved by minimizing the CAC loss can limit the network's ability to learn discriminative features (Miller et al., 2021), which is critical for achieving acceptable OSR performance (Perera et al., 2020). A further limitation of existing OSR methods relates to their application to a single data modality, i.e., RGB images, whereas OSR performance with other data modalities, such as lidar, has not been evaluated in previous work.

To overcome the limitations of existing OSR methods, we propose a new contrastive learning framework which can effectively distinguish unknown objects from known classes in different data modalities. We achieve intra-class compactness and inter-class sparsity in the feature space by training a feature encoder through contrastive learning while minimizing the class anchor clustering loss (Miller et al., 2021). During inference, the model rejects unknown classes by thresholding the virtual logit score (Wang et al., 2022). Fig. 1 illustrates the proposed Supervised Contrastive Class Anchor (S2CA) learning approach to the OSR task. We evaluate the proposed S2CA approach with a wide range of image-based and lidar-based encoder networks, and compare its performance with state-of-the-art open-set recognition and out-of-distribution (OOD) detection methods on two widely used datasets of KITTI (Geiger et al., 2013) and nuScenes (Caesar et al., 2019).

Our paper makes the following contributions:

- We propose a novel OSR method based on contrastive class anchor learning which enables effective rejection of unknown classes by increasing intra-class compactness and inter-class sparsity of known samples.

- We demonstrate that the proposed method can be applied to different data modalities and encoder networks including image-based and lidar-based networks.

- We present a thorough performance evaluation and comparison with a wide range of OSR methods, which shows the superior performance of the proposed S2CA method.

## 2 Related Work

### 2.1 Open-set Recognition

Open-set recognition has long been an attractive topic in the computer vision community. Early studies (Scheirer et al., 2013; 2014; Jain et al., 2014; Scherreik & Rigling, 2016) often involve traditional machine learning models, such as SVM, and use the probabilistic methods to threshold the prediction score.

OpenMax (Bendale & Boult, 2015) is one of the first OSR methods based on deep neural networks, which determines the center of each known class in the logit space and then creates a statistical model of the distances between correctly classified samples. It uses extreme value theory (EVT) to detect outliers by fitting a weibull function to the tail of the distance distribution. Many subsequent works involve generative models. G-OpenMax (Ge et al., 2017) trains a Generative Adversarial Network (GAN) to synthesize open-set samples and uses these to estimate open-set class activations. Neal et al. (2018) propose a data augmentation technique called counterfacutal image generation. This approach uses GANs to generate images that are visually similar to known ones, but are actually from unknown classes. The network is trained to have low confidence scores for these "known unknowns". Kong & Ramanan (2021) propose OpenGAN, which is suitable for both image classification and semantic segmentation. Generative models can do more than synthesize known unknown classes. In (Oza & Patel, 2019), the auto-encoder only learns feature representations from known classes, which leads to higher reconstruction errors for unknown classes. Yoshihashi et al. (2019) integrate classification and input reconstruction in model training, which enhances the learned feature representation. Perera et al. (2020) combine the auto-encoder with a multi-task classifier trained by minimizing a self-supervision loss and a classification loss. The self-supervision learning branch incorporates random transformations to improve the descriptiveness of feature representations. Miller et al. (2021) propose a simple OSR method, which forces known classes to form tight clusters around class anchors in logit space and reject unknown objects based on their distance to known classes. While this model is easier to train than the previous generative models, anchoring class centers at equal distances can restrict the learning of semantically meaningful features (Miller et al., 2021).

### 2.2 Out-of-distribution Detection

Detection of out-of-distribution (OOD) samples is a concept closely related to OSR, and the two terms are sometimes used interchangeably. However, OOD detection is slightly different from OSR in that OOD

samples can be completed unrelated to the training set, whereas OSR aims to reject valid but unknown classes (Gillert & von Lukas, 2021).

OOD detection methods usually map the input $x$ onto a single scalar that indicates OOD-ness. Assuming that the network is more confident about known objects, the OOD-ness leads to softmax scores that are higher for known classes than the unknown ones, making it a baseline OOD detection method (Hendrycks & Gimpel, 2017). Unknown objects tend to have a uniform softmax distribution, leading to higher entropy (Chan et al., 2021). Liang et al. (2018) propose ODIN, which adds small perturbations to the input thereby increasing the softmax score gap between known and unknown objects. Hendrycks et al. (2022) use the maximum logit as the OOD score, and show that it outperms the softmax baseline (Hendrycks & Gimpel, 2017). Kullback–Leibler divergence between the softmax probabilities and class templates has also been used as the OOD score (Hendrycks et al., 2022). Liu et al. (2020) measure OOD-ness of objects based on the Helmholtz free energy of the network output. Lee et al. (2018) use the Mahalanobis distance (MD) between the feature vector and the closest class-conditional Gaussian distribution to measure OOD-ness. Wang et al. (2022) propose Virtual-logit Matching (ViM), which combines class-agnostic information in the feature space and the class-dependent information in the logit space for OOD detection.

### 2.3 Contrastive Learning

Contrastive learning aims to learn a representation of data such that similar instances are close together in the representation space, while dissimilar instances are far apart. He et al. (2020) propose the Momentum Contrast (MoCo) for large-scale unsupervised learning, which converts the image-based contrastive learning into a dictionary look-up problem. Chen et al. (2020) propose SimCLR, which maximizes the similarity between augmented versions of the same image (positive pairs) and minimizes the similarity between different images (negative pairs). SimCLR employs an MLP projection head after the encoder network, which introduces non-linear transformations to the feature space. It also incorporates a normalization step after the projection head, which ensures that the representations have unit length, facilitating the use of cosine similarity to measure similarity between pairs. Khosla et al. (2020) propose Supervised Contrastive Learning (SCL), which extends SimCLR (Chen et al., 2020) to fully supervised scenarios. SCL contrasts all samples from the same class as positives against the negatives from the remainder of the batch, thereby forming tight class clusters in the feature space. Contrastive learning methods (He et al., 2020; Chen et al., 2020; Khosla et al., 2020) usually train an encoder with contrastive loss first, then freeze the encoder and train a linear classifier. Wang et al. (2021) argue that this two-stage learning may not be the most effective approach in a fully supervised setting, as it can reduce the compatibility between the feature encoder and the classifier. To address this issue, Wang et al. (2021) propose a hybrid framework that allows simultaneous learning of features and classifiers for long-tail image classification. Ming et al. (2023) show that contrastive learning reduces intra-class sparity and improves inter-class repulsion in the feature space, which benefits distance-based OOD detection.

## 3 Proposed Method

Given the various data modalities and the different model architectures used for object recognition in driving scenes, we propose to perform OSR in feature space. This makes our method efficient and applicable to different data modalities and encoder networks.

Fig. 2 shows an overview of the proposed S2CA framework. We train a feature encoder with Contrastive Class Anchor learning, which combines Supervised Contrastive Learning (SCL) (Khosla et al., 2020) and Class Anchor Clustering (CAC) (Miller et al., 2021). Contrastive Class Anchor learning aims to train the encoder so that the known classes show intra-class compactness and inter-class sparsity in the feature space. In this way, feature representations of known and unknown classes are separated, allowing us to model the distribution of known classes and reject unknown classes.

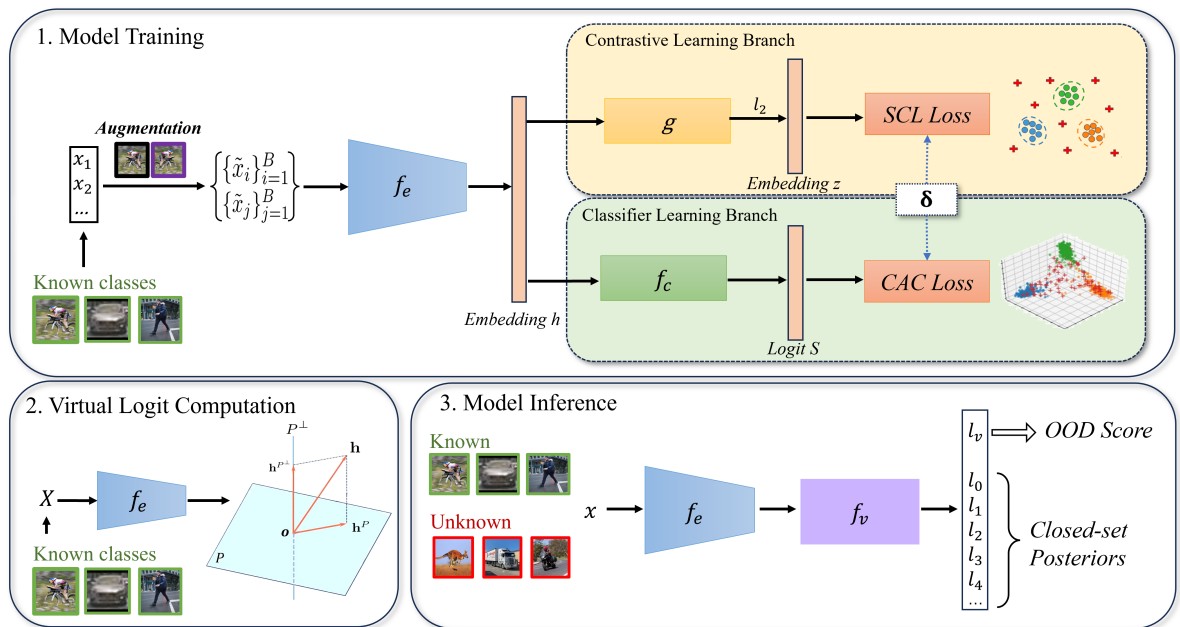

Figure 2. Overview of the proposed S2CA framework. We use a shared encoder $f_e$ to extract feature embeddings from the closed set and feed these to a supervised contrastive learning branch (SCL) and a CAC loss based classifier learning branch (step 1). The contrastive learning branch contains a non-linear MLP $g$ combined with $l_2$-normalization, which translate the feature embeddings for contrastive loss. The classifier learning branch is a single linear layer $f_c$, trained by the CAC loss. SCL is a batch contrastive approach. For each instance, SCL contrasts all instances (including the augmented instance) from the same class as positives against negatives from the remainder of the batch. Therefore, we create augmented pairs for a batch of training samples $x$ of size $B$ from the known classes, and the subsequent embeddings are also in pairs. A curriculum is designed to control the weight of these two branches, i.e., $\delta$ and $1 - \delta$, during model training. Once training is complete, we compute the distribution of virtual logits $l_v$ (step 2) based on the projection of embeddings $\|\mathbf{h}^{\mathbf{P}^\perp}\|$ which indicate OOD-ness. During model inference (step 3), we replace $f_c$ with a $N + 1$ classifier $f_v$, which produces closed-set posteriors and a virtual logit $l_v$. We threshold the virtual logit-based OOD score to reject unknown samples and classify the known ones using the normal logits.

## 3.1    Model Training

S2CA consists of a contrastive learning branch and a CAC loss based classifier learning branch. They share the same encoder $f_e$ but optimize different objective functions. The two branches together constitute Contrastive Class Anchor Learning.

### 3.1.1    Contrastive Learning Branch

Supervised Contrastive Learning (SCL) (Khosla et al., 2020) encourages objects of the same class to be tightly clustered in the feature space, while clusters belonging to different classes are far apart. These characteristics are beneficial to the OSR task. The intra-class compactness of known classes helps maintain a robust recognition performance for known objects, while the inter-class sparsity makes it easier to reject unknown objects based on their distance to the known classes.

Let $x$ and $y$ denote the input instance and the corresponding label, with $y \in N$ known classes. For each instance, S2CA creates two augmented views, $\tilde{x}_i$ and $\tilde{x}_j$, of $x$ from a predefined family $\mathcal{T}$. For image input, augmentation may include random cropping, random flipping and color jittering (for range images, we only use random flipping). For lidar point clouds, we apply geometric transformations such as random translation, random rotation and random scaling. For a mini-batch of size $B$, the corresponding augmentations are denoted as $\{\tilde{x}_i\}_{i=1}^{B}$ and $\{\tilde{x}_j\}_{j=1}^{B}$.

When $x$ is passed to the encoder network $f_e(\cdot)$, it produces the feature embedding $h = f_e(x)$. Then the MLP projector $g(\cdot)$ maps $h$ to a 128-dimensional vector. Applying $l_2$ normalization to this vector gives a normalized embedding $z$. The projector $g(\cdot)$ itself is invariant to data transformation, while it helps the encoder $f_e(\cdot)$ preserve richer information for downstream tasks (Chen et al., 2020). The SCL loss (Khosla et al., 2020) is used to maximize the agreement between $\{\tilde{z}_i\}_{i=1}^B$ and $\{\tilde{z}_j\}_{j=1}^B$ for instances of the same class. For each vector $z_i$, the SCL loss is:

$$\mathcal{L}_i^{sup} = \frac{-1}{\left|\{z_i^+\}\right|} \sum_{z_j \in \{z_i^+\}} \log \frac{\exp\left(z_i \cdot z_j / \tau\right)}{\sum_{k \neq i} \exp\left(z_i \cdot z_k / \tau\right)} \tag{1}$$

where $\{z_i^+\}$ represents all instances in the batch that have the same label $\tilde{y}_i$, $\left|\{z_i^+\}\right|$ is the number of instances in $\{z_i^+\}$, and $\tau$ is a temperature scaling parameter.

Accordingly, the loss for each mini-batch is defined as:

$$\mathcal{L}_{SCL} = \sum_{i=1}^{2B} \mathcal{L}_i^{sup} \tag{2}$$

### 3.1.2 Classifier Learning Branch

In the classifier learning branch, the feature embedding $h$ is passed to a linear classification layer $f_c$ to produce the N-class logit vector $s$. The CAC loss (Miller et al., 2021) is defined based on the distance between the logit vectors $s$ and a set of anchored class centers $C = (c_1, \cdots, c_n)$ with a predefined magnitude $\alpha$ set equally for all classes. Let $d$ denote a vector of Euclidean distances between $s$ and each anchored class center in $C$, and $f(\cdot)$ denote the classification function, the CAC loss is written as:

$$\mathcal{L}_{CAC} = log(1 + \sum_{j \neq y}^B e^{d_y - d_j}) + \lambda \left\| f(x) - c_y \right\|_2 \tag{3}$$

where $\lambda$ is a hyperparameter to balance the two loss terms.

The first term of $\mathcal{L}_{CAC}$ is modified from the tuplet loss (Sohn, 2016). It forces $s$ to maximize the difference between the distance to the correct class center and the distances to all other class centers. The second term minimizes the distance between $s$ and the correct class center.

### 3.1.3 Transition from Contrastive Learning to Classifier Learning

To ensure compatibility between the feature encoder and the classifier, we follow a curriculum strategy (Zhou et al., 2020) to adjust the weighting of the two branches. The model will first focus on supervised contrastive learning (Khosla et al., 2020) and then smoothly transitions to classifier learning as the training progresses. The final objective function is the combination of Eq. (2) and Eq. (3).

$$\mathcal{L}_{S2CA} = \delta \mathcal{L}_{SCL} + (1 - \delta)\mathcal{L}_{CAC} \tag{4}$$

Note that $\mathcal{L}_{CAC}$ in Eq. 4 is the sum of each individual loss in the mini-batch (like the batch form of the SCL loss shown in Eq. 2). $\delta$ is a parameter that decreases as the training progresses. For each epoch $t_i$, $\delta = 1 - \frac{t_i}{t_{max}}$ where $t_{max}$ is the maximum epoch number. Initially, $\delta$ is large, and the model focuses on optimizing the SCL loss to learn semantically meaningful features. As the training progresses, $\delta$ is reduced and the model smoothly shifts its focus to the CAC loss, which fine-tunes the model from a global perspective.

The combination of SCL and CAC has several advantages. While using the CAC loss alone can restrict the discriminative power of the learned features (Miller et al., 2021), the SCL helps the encoder learn a richer feature space. The SCL also benefits from the CAC. As a batch contrastive algorithm, SCL only contrasts samples in the same mini-batch. To achieve better performance, SCL requires larger batch sizes or more

training epochs, which increases the computational complexity of the training (Khosla et al., 2020). The CAC mitigates this by comparing each individual sample with its corresponding class anchor. In each epoch, all training instances are pulled to the corresponding class anchors to form tight clusters. This results in faster convergence of contrastive learning with smaller batch sizes.

## 3.2 Model Inference

For model inference, we use virtual logit (Wang et al., 2022) to extract information in the feature space and combine it with ordinary logit to form the OOD score and reject unknown objects. Unlike the distance-based rejection score used by Miller et al. (2021), which ignores the class-agnostic information in the feature space, the virtual logit matching (ViM) combines the class-agnostic information in the feature space with the class-dependent information to reject unknown objects and classify the known ones.

### 3.2.1 Virtual Logit

The virtual logit is the scaled norm of the projection of the feature vector $h$ to the orthogonal complement of the principal subspace of the feature space. To simplify the subsequent logit computation, we offset $h$ with a new origin $o = -(W^T)^+b$, where $W$ is the weight in $f_c$, $b$ is the bias and $(\cdot)^+$ is the Moore-Penrose inverse. This transforms the feature vectors to a new bias-free coordinate system. To distinguish these from the previous features $h$, the feature vectors in the new coordinate system are denoted $\mathbf{h}$. We further denote the feature matrix of the training set as $\mathbf{H}$.

The principal subspace $P$ of the feature space is defined by the eigen decomposition of $\mathbf{H}^T\mathbf{H}$:

$$\mathbf{H}^T\mathbf{H} = Q\Lambda Q^{-1} \tag{5}$$

where $\Lambda$ contains the eigenvalues in descending order, and the first $D$ columns of $Q$ define the D-dimensional principal subspace $P$. Let $P^\perp$ be the orthogonal complement of $P$, and $\mathbf{h}^{P^\perp}$ the projection of $\mathbf{h}$ to $P^\perp$. After this projection, known data tend to have a smaller norm, whereas unknown data exhibit a larger norm. This facilitates the differentiation between known and unknown samples. The virtual logit is then obtained as (Wang et al., 2022):

$$l_v = \beta\|\mathbf{h}^{P^\perp}\| \tag{6}$$

where $\beta$ is the scaling constant:

$$\beta = \frac{\sum_{i=1}^K max_{j=1,\dots,N}\{l_j^i\}}{\sum_{i=1}^K \|\mathbf{h}_i^{P^\perp}\|} \tag{7}$$

where $N$ is the number of known classes, $K$ is the number of training samples, and $l_j^i$ is the $j-$th logit of $\mathbf{h}_i$.

Following Wang et al. (2022), we obtain the ViM score by appending the energy score (Liu et al., 2020) to the virtual logit:

$$v = \beta\|\mathbf{h}^{P^\perp}\| - \ln\sum_{i=1}^N e^{l_i} \tag{8}$$

which has a better performance than each individual score.

### 3.2.2 Inference Pipeline

During inference, the final classification layer $f_c$ is replaced by a new module $f_v$, which produces a logit vector of length $N+1$, that is $N$ normal logits and one virtual logit. $f_v$ is able to invoke both the encoder $f_e$ and the classification layer $f_c$. It uses the encoder $f_e$ to extract the feature matrix $H$ from the training set $X$, project $H$ into a bias-free coordinate system and calculate the $D$ dimensional principal subspace $P$. We

Table 1. Data Modalities and Encoder Networks

| Input Type | Network | Architecture |
|---|---|---|
| RGB Image | WRN-40-2 | Wide ResNet (Zagoruyko & Komodakis, 2016) |
| Range Image | WRN-40-2 | Wide ResNet (Zagoruyko & Komodakis, 2016) |
| 3D Voxel Grid | VoxNet | Volumetric CNN (Maturana & Scherer, 2015) |
| 3D Voxel Grid | LightNet | Volumetric CNN (Zhi et al., 2018) |
| Point Cloud | DGCNN | Dynamic Graph CNN (Wang et al., 2019) |
| Point Cloud | PCT | Point Transformer (Guo et al., 2021) |

calculate the scaling variable $\beta$ based on Eq. (7). For normal logits, $f_v$ directly passes the feature embedding to $f_c$.

Now, we have all the parameters to compute the OOD score $v$ (Eq. (8)). A larger value for $v$ indicates that the object is more likely to be unknown. As such, unknown objects can be rejected by setting a threshold $\theta$. The decision rule is as follows:

$$prediction = \begin{cases} class \ i = argmax(l), & v \leq \theta \\ unknown, & v > \theta \end{cases} \tag{9}$$

The threshold $\theta$ can be determined without requiring auxiliary unknown data by analyzing the distribution of OOD scores of the known samples in the training set.

### 3.3 Implementation Details

For training, we use the Cosine Annealing Learning Rate (Loshchilov & Hutter, 2017), with a maximum learning rate of 0.1 and a minimum of $10^{-4}$. In the initial epochs, the learning rate decreases slowly to speed up the contrastive learning process. The learning rate begins to drop rapidly midway through the training to facilitate classifier learning. Towards the end of the training, the learning rate becomes very small, so that the classifier can be fine-tuned. The training takes up to 200 epochs, and is optimized by stochastic gradient descent (SGD) with a momentum of 0.9. For the SCL loss, the temperature is set to 0.1 and the batch size is limited to 256. For the CAC loss, the anchor magnitude $\alpha$ is set to 10 and the weight variable $\lambda$ is set to 0.3. For the virtual logit, the dimensionality of the principal subspace is set to 64.

## 4 Experiments

### 4.1 Comparison with State-of-the-art Methods

We compare the performance of S2CA with existing OSR and OOD detection methods, for which source code is available from PyTorch-OOD library (Kirchheim et al., 2022) or GitHub. Specifically, we select SoftMax (Hendrycks & Gimpel, 2017), ODIN (Liang et al., 2018), MD (Lee et al., 2018), Energy (Liu et al., 2020), Entropy (Chan et al., 2021), MaxLogit (Hendrycks et al., 2022), KL-M (Hendrycks et al., 2022), ViM (Wang et al., 2022), and CAC (Miller et al., 2021). The selected methods are compatible with different data modalities and encoder networks.

### 4.2 Encoder Networks

To evaluate the open-set recognition performance with both image and lidar data several encoder networks were used as shown in Tab. 1. For RGB image data, we use a 40-layer wide residual network with a widening factor 2 (WRN-40-2) (Zagoruyko & Komodakis, 2016) for feature extraction. For lidar data, we use encoder networks operating on raw point clouds, range images, and 3D voxel grids. For range images, we use the same Wide ResNet as for RGB image data. For lidar data represented as 3D voxel grids, we use VoxNet (Maturana & Scherer, 2015) and LightNet (Zhi et al., 2018) as the encoder networks. For lidar point clouds, we use

DGCNN (Wang et al., 2019) and Point Cloud Transformer (PCT) (Guo et al., 2021), which directly take point clouds as input.

## 4.3 Datasets

We use two public autonomous driving datasets for evaluation: nuScenes (Caesar et al., 2019) and KITTI (Geiger et al., 2013). For the object detection task, nuScenes has 23 classes, while KITTI (Geiger et al., 2013) has 8 classes. Each dataset was divided into three parts: $x_{train}$, $x_{test}$ and $x_{OOD}$. $x_{train}$ and $x_{test}$ contained samples of known classes for training and testing. $x_{OOD}$ refers to unknown classes (out of distribution). We selected the classes with the fewest annotations as unknown. In nuScenes, unknown classes were *stroller*, *personal mobility vehicle*, *animal*, and *ambulance*. In KITTI, unknown classes were *truck*, *misc* (miscellaneous objects, such as trailers and segways), *tram*, and *sitting person*. We first located each object using the given bounding box and then extracted these from the image and lidar point cloud data. Heavily occluded or completely invisible objects were filtered out. Following Neal et al. (2018), we calculate the *openness* of our datasets according to: $openness = 1 - \sqrt{\frac{K}{K+U}}$, where $K$ is the number of known classes and $U$ is the number of unknown classes. Accordingly, the *openness* of our nuScenes dataset was found to be 9.1%, while the *openness* of KITTI was 29.3%.

## 4.4 Evaluation Metrics

The effectiveness of an open set recognition model can be evaluated by its overall classification accuracy or F-score when detecting unknown classes in mixed data. However, these combined metrics are affected not only by the quality of the OSR model, but also by the calibration parameters chosen (Neal et al., 2018). As a result, we evaluate the performance of OSR models using two metrics: Area Under the ROC Curve (AUROC) and Closed Set Accuracy.

**Area Under the ROC Curve (AUROC)**  AUROC uses all possible thresholds to plot an ROC curve of the True Postive Rate (TPR) against False Positive Rate (FPR). Ideally, known objects should have a lower OOD-ness score, whereas unknown objects should have a higher score. If all unknown classes have higher scores than the known ones, the TPR will be 1 and the FPR will be 0 after trying all thresholds, resulting in a perfect AUROC value of 1. Lower AUROC values indicate that unknown objects are mixed with the known ones.

**Closed-set Accuracy**  An open-set classifier must maintain its capacity for recognizing closed-set objects as well. As a result, we will further evaluate the classification accuracy of known classes.

# 5 Results and Discussion

## 5.1 Open-set Recognition Performance

Tab. 2 shows the performance of the OSR methods with different encoder networks as measured by AUROC. It can be seen that overall S2CA achieves leading performance on both datasets across different data modality and encoder networks. Interestingly, the OSR methods perform well with lidar data when paired with suitable encoder networks. Point-based encoder networks generally outperform projection-based and voxel-based models. The range image representation of lidar data does not seem to benefit the open-set recognition performance, especially when the lidar point cloud is sparse. This can be seen from the performance of the range image-based models on nuScenes, which is noticeably worse than on KITTI. This is because nuScenes lidar data are captured by a 32-channel lidar, so the point cloud is relatively sparse and the range images have a lower resolution as compared to KITTI which uses a 64-channel lidar. Notably, ODIN (Liang et al., 2018) and MD (Lee et al., 2018) perform quite poorly on nuScenes range images. This can be attributed to the addition of noise during preprocessing step in these methods, which has a negative impact on the recognition performance. The comparable performance of the OSR methods with different image-based and lidar-based encoder networks implies that the power of open-set recognition lies in the feature learning. The modality of the data is not as crucial as the information it carries.

Table 2. Performance of S2CA compared with state of the art OSR methods. Open-set recognition performance is measured using AUROC (%). For each encoder network, the best and second-best OSR results are emphasized.

| Dataset | OSR Method | Trainable | WRN (RGB) | WRN (Range) | VoxNet (Voxel) | LightNet (Voxel) | DGCNN (Point) | PCT (Point) |
|---|---|---|---|---|---|---|---|---|
| nuScenes | SoftMax (Hendrycks & Gimpel, 2017) | | 79.26 | 60.52 | 71.78 | 69.17 | 80.50 | 80.92 |
| | ODIN (Liang et al., 2018) | | 73.44 | 34.30 | 71.70 | 69.30 | 81.16 | 80.20 |
| | MD (Lee et al., 2018) | | 78.07 | 23.42 | 74.43 | 72.02 | 78.94 | 80.51 |
| | Energy (Liu et al., 2020) | ✓ | 79.72 | 62.53 | 73.93 | 71.11 | **81.73** | 80.82 |
| | Entropy (Chan et al., 2021) | ✓ | 79.56 | 61.80 | 73.05 | 70.27 | 81.14 | 81.09 |
| | MaxLogit (Hendrycks et al., 2022) | | 79.65 | 62.04 | 73.20 | 70.36 | 81.45 | 81.25 |
| | KL-M (Hendrycks et al., 2022) | | 78.42 | 52.58 | 56.87 | 52.93 | 61.69 | 67.52 |
| | ViM (Wang et al., 2022) | | 80.73 | **70.59** | **75.53** | **74.59** | 80.41 | 82.28 |
| | CAC (Miller et al., 2021) | ✓ | **85.32** | 69.63 | 72.40 | 74.04 | 77.65 | **82.80** |
| | S2CA (ours) | ✓ | **82.88** | **70.99** | **76.05** | **77.25** | **83.12** | **83.21** |
| KITTI | SoftMax (Hendrycks & Gimpel, 2017) | | 74.07 | 75.68 | 76.88 | 77.92 | 72.50 | 75.40 |
| | ODIN (Liang et al., 2018) | | 62.09 | 61.72 | 76.98 | 77.90 | 70.85 | 75.31 |
| | MD (Lee et al., 2018) | | 79.78 | 61.23 | 74.01 | 76.77 | **86.90** | **78.19** |
| | Energy (Liu et al., 2020) | ✓ | 74.91 | 72.10 | 76.03 | 77.49 | 72.89 | 74.88 |
| | Entropy (Chan et al., 2021) | ✓ | 74.08 | 75.66 | **77.01** | **78.29** | 72.60 | 75.27 |
| | MaxLogit (Hendrycks et al., 2022) | | 74.90 | 71.99 | 76.07 | 77.48 | 72.82 | 74.93 |
| | KL-M (Hendrycks et al., 2022) | | 62.55 | 69.15 | 61.70 | 60.47 | 72.88 | 75.33 |
| | ViM (Wang et al., 2022) | | **80.38** | **76.56** | 76.68 | 77.41 | 85.27 | 77.39 |
| | CAC (Miller et al., 2021) | ✓ | 73.71 | 72.12 | 74.44 | 77.58 | 71.83 | 74.39 |
| | S2CA (ours) | ✓ | **81.23** | **79.64** | **77.39** | **79.28** | **87.48** | **81.94** |

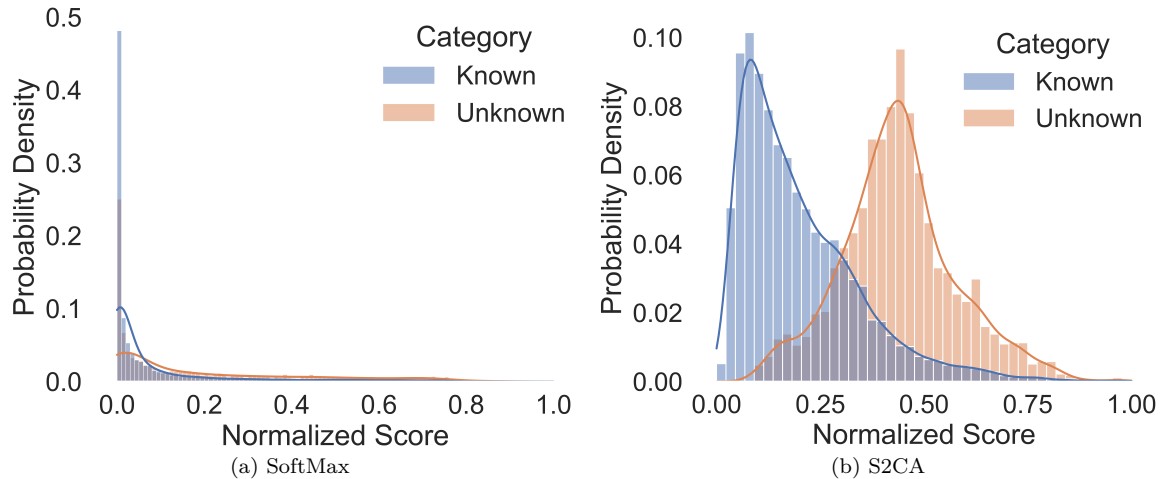

(a) SoftMax    (b) S2CA

Figure 3. Distribution of OOD scores for known and unknown samples obtained by the Softmax (inverted) baseline (a), and the proposed S2CA (b). For fair comparison, all scores are normalized to [0,1].

The superior performance of the proposed S2CA can be attributed to its ability to learn discriminative features, which results in OOD scores that are higher for unknown classes. Fig. 3 shows the probability density distributions of the normalized OOD scores for both known and unknown classes in the KITTI dataset. For the SoftMax baseline (Hendrycks & Gimpel, 2017) in Fig. 3a, both distributions exhibit a long-tailed pattern and there is a high degree of overlap between them, whereas S2CA ( Fig. 3b) presents a noticeable difference between the two distributions, making it easier to determine an appropriate threshold to distinguish the known and unknown classes effectively. The significant difference between the two figures explains the superior performance of S2CA in the OSR task.

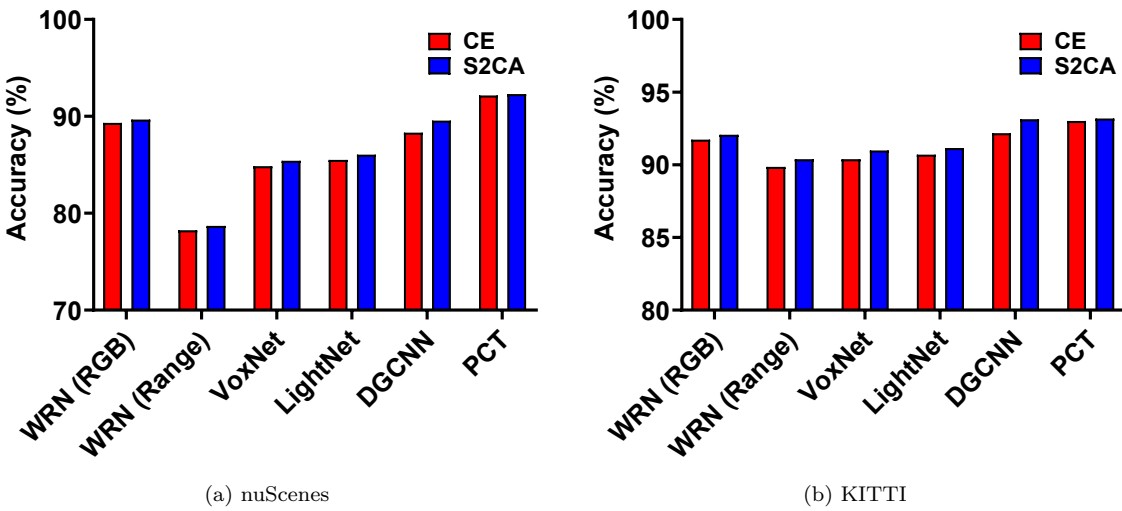

(a) nuScenes

(b) KITTI

Figure 4. Closed-set accuracy of S2CA with different encoder networks compared with the cross-entropy baseline on nuScenes (a) and KITTI (b).

Fig. 4 shows the closed-set accuracy of S2CA as compared with the conventional cross-entropy baseline. It can be seen that for all encoder networks the S2CA framework maintains a comparable closed-set performance as the conventional cross-entropy baseline. This again indicates that the ability of S2CA to extract rich features from the data is not only useful for rejecting unknown objects, but also helps maintain a high classification accuracy on known classes.

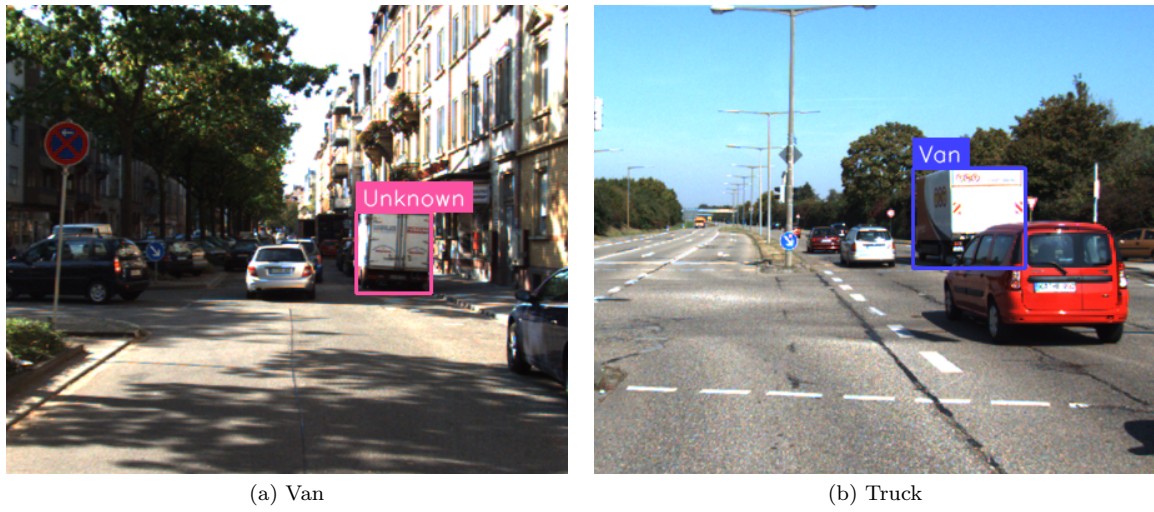

(a) Van

(b) Truck

Figure 5. Examples of failure cases. Van (a) is a known class, while Truck (b) is unknown.

A close examination of the errors reveals that most misclassified samples are objects that are similar in appearence and are too far from the sensor. Fig. 5 shows an example of a known (van) and an unknown object (truck) which were misclassified by S2CA. In such cases, where the unknown object looks very similar to a known class, rejecting the unknown objects and correctly classifying the known ones is challenging.

## 5.2 Ablation Study

The S2CA framework consists of three important components: contrastive learning, class anchor clustering, and virtual logit matching. To evaluate the contribution of each individual component to the final perfor-

mance, we systematically remove these components from the S2CA framework to create a set of baseline models:

**CLS** - A standard N class classifier, denoted $f$, trained with cross-entropy loss. Unknown objects are identified using the SoftMax score.

**CLS + ViM (CLS V.)** - This baseline uses the same classifier $f$, but rejects unknown objects based on the ViM (Wang et al., 2022) score.

**CLS + CAC + ViM (C.C.V)** - The classifier $f$ is trained with CAC (Miller et al., 2021) loss, but rejects unknown objects based on ViM (Wang et al., 2022) score.

**Contrastive CLS (Cont. CLS)** - We further add the supervised contrastive learning (SCL) (Khosla et al., 2020) branch to $f$. The model rejects unknown objects based on the ViM (Wang et al., 2022) score.

**S2CA** - The complete S2CA framework.

Tab. 3 illustrates the results of the ablation study. The complete S2CA framework with different encoder networks consistently outperforms all competing baseline models in terms of AUROC. It is evident that ViM is a key contributor to open-set performance. We can see a substantial improvement in AUROC between CLS and CLS V. The SoftMax score in CLS depends solely on the information in the logit space, and OOD samples tend to have smaller SoftMax probabilities than in-distribution samples. In contrast, ViM incorporates class-agnostic information in the feature space and class-dependent information in the logit space, which better reflects OOD-ness (Wang et al., 2022). However, ViM neglects the variance of maximum logits in the calculation of the scaling parameter per model, weakening the importance of information in the feature space in the final decision-making process. This limitation is effectively addressed by CAC loss, which forces known objects to form tight clusters in logit space. Our results show that CAC alone provides limited improvements in open-set performance, with AUROC scores of CLS V. and C.C.V showing small differences. We can also observe that contrastive learning leads to overall improvements in AUROC. By incorporating supervised contrastive learning, Cont. CLS surpasses the performance of both CLS V. and C.C.V. Furthermore, the complete S2CA framework also uses CAC loss to mitigate overconfident classification and address the limitations associated with the ViM score. The cumulative effect of these components makes the S2CA framework superior to all baseline methods, underscoring its potential as a comprehensive and effective approach to open-set recognition.

Table 3. Performance of the complete S2CA framework in terms of AUROC compared to other baselines

| Dataset | Baseline | Encoder Network | | | | | |
| | | WRN (RGB) | WRN (Range) | VoxNet (Voxel) | LightNet (Voxel) | DGCNN (Point) | PCT (Point) |
|---|---|---|---|---|---|---|---|
| nuScenes | CLS | 79.26 | 60.52 | 71.78 | 69.17 | 80.5 | 80.92 |
| | CLS V. | 80.73 | 70.59 | 75.53 | 74.55 | 80.41 | 82.28 |
| | C.C.V. | 80.32 | 70.63 | 75.4 | 74.04 | 80.43 | 82.8 |
| | Cont. CLS | 81.59 | 70.89 | 75.8 | 76.12 | 82.53 | 82.87 |
| | S2CA | **82.88** | **70.99** | **76.05** | **77.25** | **83.12** | **83.21** |
| KITTI | CLS | 74.07 | 75.68 | 76.88 | 77.92 | 72.5 | 75.4 |
| | CLS V. | 80.38 | 76.56 | 76.68 | 77.41 | 85.27 | 77.39 |
| | C.C.V. | 80.71 | 76.12 | 76.44 | 77.58 | 85.83 | 77.56 |
| | Cont. CLS | 80.33 | 77.58 | 77.23 | 78.56 | 86.21 | 80.04 |
| | S2CA | **81.23** | **79.64** | **77.39** | **79.28** | **87.48** | **81.94** |

Following Sun et al. (2021), we experiment with different projector dimensions and temperatures for contrastive learning. The results for the RGB image classifier trained with S2CA are shown in Tab. 4. As it can be seen, the choice of hyperparameters has a minor impact on the performance of S2CA.

Tab. 5 shows the performance of post-training OOD scores applied to the RGB image classifier trained with S2CA. ViM (Wang et al., 2022) has the best performance since it combines class-agnostic information in the feature space and the class-dependent information in the logit space for OOD detection.

| Contrast Head | Temperature ($\tau$) | | |
| --- | --- | --- | --- |
| Dimension | 0.07 | 0.2 | 0.5 |
| $D_C = 128$ | 81.23 | 80.83 | 80.70 |
| $D_C = 256$ | 80.98 | 79.12 | 79.92 |

Table 4. Effect of contrastive learning hyperparameters on OSR performance as measured by AUROC on KITTI.

| Post-training OOD score | nuScenes | KITTI |
| --- | --- | --- |
| SoftMax (Hendrycks & Gimpel, 2017) | 79.95 | 74.77 |
| MaxLogit (Hendrycks et al., 2022) | 80.50 | 75.17 |
| ODIN (Liang et al., 2018) | 72.23 | 65.49 |
| MD (Lee et al., 2018) | 78.98 | 80.59 |
| ViM (Wang et al., 2022) | **82.88** | **81.23** |

Table 5. Comparison of OOD scores applied with S2CA after training.

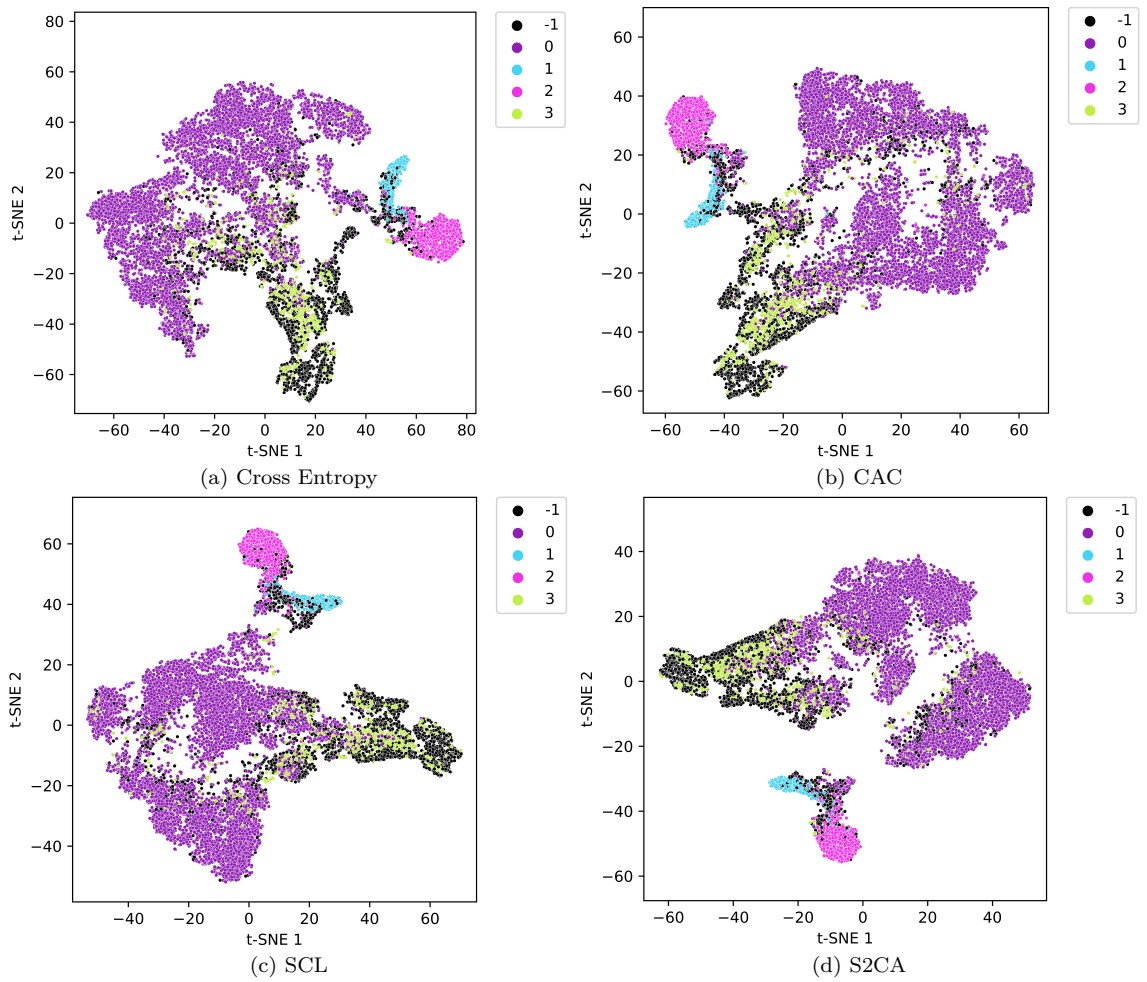

Figure 6. T-SNE visualization of feature space trained on different objectives. Numerical labels correspond to object categories: -1 denotes unknown objects, 0 represents Car, 1 represents Cyclist, 2 represents Pedestrian and 3 represents Van.

Fig. 6 shows the feature space of the image classifier trained by different objectives. Without contrastive learning (Fig. 6a and Fig. 6b), there is significant overlap between the features of known classes and OOD samples. Contrastive learning promotes the intra-class compactness, with S2CA (Fig. 6d) showing more compact clusters as compared to the original SCL (Fig. 6c). Due to the semantic similarity between OOD samples and known classes, they are still close to each other in the feature space. But contrastive learning makes OOD samples gather at the edges of the known clusters, making it easier to reject unknown samples.

Table 6. OOD detection performance of `S2CA` with a ResNet-34 backbone trained on CIFAR-100 (ID) compared with state-of-the-art methods.

| Method | SVHN | | Places365 | | OOD Dataset LSUN | | iSUN | | Texture | | Average | |
|---|---|---|---|---|---|---|---|---|---|---|---|---|
| | FPR↓ | AUROC↑ | FPR↓ | AUROC↑ | FPR↓ | AUROC↑ | FPR↓ | AUROC↑ | FPR↓ | AUROC↑ | FPR↓ | AUROC↑ |
| Without Contrastive Learning | | | | | | | | | | | | |
| MSP | 78.89 | 79.80 | 84.38 | 74.21 | 83.47 | 75.28 | 84.61 | 74.51 | 86.51 | 72.53 | 83.12 | 75.27 |
| ODIN | 70.16 | 84.88 | 82.16 | 75.19 | 76.36 | 80.10 | 79.54 | 79.16 | 85.28 | 75.23 | 78.70 | 79.11 |
| Mahalanobis | 87.09 | 80.62 | 84.63 | 73.89 | 84.15 | 79.43 | 83.18 | 78.83 | 61.72 | 84.87 | 80.15 | 79.53 |
| Energy | 66.91 | 85.25 | 81.41 | 76.37 | 59.77 | 86.69 | 66.52 | 84.49 | 79.01 | 79.96 | 70.72 | 82.55 |
| GODIN | 74.64 | 84.03 | 89.13 | 68.96 | 93.33 | 67.22 | 94.25 | 65.26 | 86.52 | 69.39 | 87.57 | 70.97 |
| LogitNorm | 59.60 | 90.74 | 80.25 | 78.58 | 81.07 | 82.99 | 84.19 | 80.77 | 86.64 | 75.60 | 78.35 | 81.74 |
| With Contrastive Learning | | | | | | | | | | | | |
| ProxyAnchor | 87.21 | 82.43 | 70.10 | 79.84 | 37.19 | 91.68 | 70.01 | 84.96 | 65.64 | 84.99 | 66.03 | 84.78 |
| CE + SimCLR | 24.82 | 94.45 | 86.63 | 71.48 | 56.40 | 89.00 | 66.52 | 83.82 | 63.74 | 82.01 | 59.62 | 84.15 |
| CSI | 44.53 | 92.65 | 79.08 | 76.27 | 75.58 | 83.78 | 76.62 | 84.98 | 61.61 | 86.47 | 67.48 | 84.83 |
| SSD+ | 31.19 | 94.19 | 77.74 | 79.90 | 79.39 | 85.18 | 80.85 | 84.08 | 66.63 | 86.18 | 67.16 | 85.90 |
| KNN+ | 39.23 | 92.78 | 80.74 | 77.58 | 48.99 | 89.30 | 74.99 | 82.69 | 57.15 | 88.35 | 60.22 | 86.14 |
| CIDER | 23.09 | 95.16 | 79.63 | 73.43 | 16.16 | 96.33 | 71.68 | 82.98 | 43.87 | 90.42 | **46.89** | 87.67 |
| S2CA | 27.08 | 94.56 | 69.90 | 80.87 | 40.91 | 92.41 | 67.68 | 84.34 | 49.04 | 88.99 | 50.92 | **88.23** |

## 5.3 Evaluation on Out-of-Distribution Benchmarks

Our results generally support the hypothesis that the combination of contrastive learning and class anchor clustering provides a convincing solution to the problem of open set recognition in driving scenes. Similar findings have been reported for general OOD benchmarks by another recent work, CIDER (Ming et al., 2023), which also adopts the concept of supervised contrastive learning and class anchor (or prototype).

To evaluate the performance of S2CA on OOD benchmarks, we follow the experimental setup of CIDER (Ming et al., 2023), and use CIFAR100 (Krizhevsky et al., 2009) as the in-distribution dataset to train S2CA with a ResNet-34 backbone (He et al., 2016). OOD test datasets include SVHN (Netzer et al., 2011), Places365 (Zhou et al., 2017), Textures (Cimpoi et al., 2014), LSUN (Yu et al., 2015), and iSUN (Xu et al., 2015). As shown in Tab. 6, S2CA achieves similar results to CIDER, with a slightly higher overall AUROC and a slightly worse FPR. When compared to other contrastive learning-based methods, such as ProxyAnchor (Kim et al., 2020), CSI (Tack et al., 2020), SSD+ (Sehwag et al., 2021), and KNN+ (Sun et al., 2022), S2CA demonstrates the best overall performance on both metrics.

While both CIDER and S2CA use contrastive learning and class anchors, they have subtle differences. S2CA places fixed class anchors in the logit space, as the ground truth label of known classes, whereas CIDER estimates the prototypes (anchors) directly from sample embeddings using EMA in the feature space. Furthermore, S2CA jointly optimizes SCL and CAC loss. In contrast, CIDER uses maximum likelihood estimation (MLE) loss and dispersion loss to promote intra-class compactness and inter-class sparsity in the feature space. Also S2CA uses the virtual logit as OOD score, taking advantage of information from both feature space and logit space, whereas CIDER adopts k-NN cosine similarity as the OOD score, which only uses information in the feature space.

## 5.4 Theoretical Analysis

Softmax probability as a baseline OOD detection method achieves modest success, because the neural network is less confident with unknown input. This observation supports previous results, such as (Hendrycks & Gimpel, 2017) and (Kang et al., 2024), which also show that the predictions of the neural network converge to a constant value when inputs become more OOD. However, as shown by Pearce et al. (2021), since neural

networks are not bijective, the features of in-distribution input and OOD input in the final-layer can be the same, resulting in the failure of softmax probability in rejecting unknown samples. The network can also produce lower softmax confidence for corrupted known samples. Since OOD data are absent in the training process, neural networks need to learn robust features of known classes to reduce overlap with OOD features. Contrastive learning promotes the alignment of positive pairs and uniformity of the feature space (Wang & Isola, 2020). In a fully supervised setting, where positive pairs are defined by class label (Khosla et al., 2020), the alignment of positive pairs encourages known classes to form tight clusters, thus improving invariance to noise, while the uniformity helps preserve as much information from the training samples as possible, but also encourages inter-class sparsity. As OOD samples lie between ID clusters, optimizing a large distance among ID clusters benefits OOD detection (Ming et al., 2023). Traditional contrastive learning methods such as SCL (Khosla et al., 2020) usually train an encoder with contrastive loss first, then freeze the encoder and train a linear classifier. As shown by Wang et al. (2021), this two-stage learning may not be the most effective approach in a fully supervised setting, as it can reduce the compatibility between the feature encoder and the classifier. S2CA jointly optimizes the model with feature-based SCL loss and logit-based CAC loss (Miller et al., 2021). Contrastive learning helps the encoder learn a richer feature space invariant to noise, while anchor loss will penalize the network when it makes low-confidence or wrong predictions. As driving scenes are characterized by varying weather and lighting condition (Michaelis et al., 2019), the S2CA framework enables the model to learn robust feature embeddings from noisy data, improving the classification of known classes and rejection of unknown ones.

## 6 Conclusion

We proposed an effective and adaptive OSR method based on contrastive class anchor learning. Our experimental results show that the proposed method achieves state-of-the-art performance with different data modalities and encoder networks. Future work will investigate failure detection, which focuses on detecting misclassification, covariate shifts (corruption shift and domain shift), and new class shifts (Jaeger et al., 2023). In the future work, we will work on a unified failure detection method, which detects both misclassified known samples and out-of-distribution samples under various domains. For real-world application, S2CA classifier can be integrated into a two-step object detector to make a complete open-set object detection pipeline, for different data modalities. For image input, S2CA can be applied to methods such as VOS (Du et al., 2022), which appends open-set classifier after Region Proposal Network (RPN) (Ren et al., 2015). For LiDAR input, object proposals can be obtained via ground segmentation and clustering (Nunes et al., 2022).

## Acknowledgement

This research was supported by The University of Melbourne's Research Computing Services and the Petascale Campus Initiative. The first author acknowledges the financial support from the University of Melbourne through the Melbourne Research Scholarship.

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
