# OpenReview forum: "Contrastive Class Anchor Learning for Open Set Object Recognition in Driving Scenes"
_TMLR — Accepted by TMLR_

### Review · Reviewer_gKuR · 2024-08-27

**Summary Of Contributions:**

1. Introduce supervised contrastive loss into open set object recognition task to improve model's ability to reject unknown sample and known sample class distribution.
2. Use different modalities data, different encoder networks, and different datasets to prove framework effectiveness.

**Audience:**

Yes

**Claims And Evidence:**

No

**Requested Changes:**

1. It would be better to provide a comprehensive illustration (theoretical derivation or experimental proof) of why contrastive learning could improve open set object detection tasks to enrich paper content.
2. The author should provide detailed analysis of the experiment, such as an ablation study on contrastive learning and t-sne visualization on features to illustrate different module effects.
3. Some visualization images are unsightly. For example, the author could reference Figure 6 in yolo[1] to improve the quality of Figure 5.

[1] Redmon, Joseph, Santosh Divvala, Ross Girshick, and Ali Farhadi. "You only look once: Unified, real-time object detection." In Proceedings of the IEEE conference on computer vision and pattern recognition, pp. 779-788. 2016.

**Strengths And Weaknesses:**

Strengths:
1. The proposed method is comprehensively explained through detailed formulas and illustrative figures.
2. The effectiveness of the method is demonstrated using a variety of encoder networks.

Weaknesses:
1. The novelty of the framework is limited, as it seems to only use contrastive loss based on the previous method.
2. Lack of theoretical derivation. Though contrastive loss improves model closed-set accuracy, it would be better to provide corresponding theoretial analysis to prove why this module works. Some related works[1][2] may help.
3. Missing some ablation study. For example, the ablation study of projector g(.). It could be better to provide corresponding ablation study like Table 6 in FSCE[3].

[1] Wang, Tongzhou, and Phillip Isola. "Understanding contrastive representation learning through alignment and uniformity on the hypersphere." In International conference on machine learning, pp. 9929-9939. PMLR, 2020.

[2] Khosla, Prannay, Piotr Teterwak, Chen Wang, Aaron Sarna, Yonglong Tian, Phillip Isola, Aaron Maschinot, Ce Liu, and Dilip Krishnan. "Supervised Contrastive Learning-Supplementary Material."

[3] Sun, Bo, Banghuai Li, Shengcai Cai, Ye Yuan, and Chi Zhang. "FSCE: Few-shot object detection via contrastive proposal encoding." In Proceedings of the IEEE/CVF conference on computer vision and pattern recognition, pp. 7352-7362. 2021.

---

> ### Author Response · Authors · 2024-09-16
> **Response to Reviewer gKuR**
>
> Thank you for your comments. Please see our responses below:
>
> 1.	We have revised the introduction section to highlight the novelty of the paper.
>
> 2.	As we have shown through experimental evaluation and ablation studies in Sections 5.1 and 5.2, the improved performance of S2CA is not the result of contrastive learning alone, but is the combined effect of contrastive learning, class anchor learning, and virtual logit matching. Our ablation results in Tables 3 and 5 clearly demonstrate how these components contribute to improved performance of S2CA.
>
> 3.	We have included additional ablations including a comparison of contrastive hyperparameters in Table 4 as you suggested. We also provide the T-SNE visualization of feature space for different modules in Figure 6.
>
> 4.	We have modified Figure 5 based on your suggestion and improved the quality of all figures.

---

> > ### Comment · Reviewer_gKuR · 2024-09-23
> > **About theoretical analysis**
> >
> > Thank you for conducting these ablation studies to demonstrate the effectiveness of each module. Could you also provide some theoretical analysis? I believe this would strengthen the overall work significantly.

---

> > > ### Comment · Reviewer_hhVc · 2024-09-23
> > >
> > > Although I agree that theoretical analysis (in the sense of mathematical proof) would additional value to the paper, I personally believe it is not *essential* for a paper in the field of OOD detection/open set recognition. Most similar work tends to be based on intuitive/empirical insights (like [1]).
> > >
> > > To supplement reviewer gKuR, [2,3] may be helpful starting points for the authors if they choose to undertake further analysis.
> > >
> > > [1] Ming et al. *How to Exploit Hyperspherical Embeddings for Out-of-Distribution Detection?* ICLR 2023
> > >
> > > [2] Pearce et al. *Understanding Softmax Confidence and Uncertainty*, 2021
> > >
> > > [3] Kang et al. *Deep Neural Networks Tend To Extrapolate Predictably*, ICLR 2024

---

> > > ### Author Response · Authors · 2024-09-25
> > > **Response to Reviewer gKuR**
> > >
> > > Thank you for your comments. We have included the theoretical analysis in Section 5.4, based on the literature suggested by you and Reviewer hhVc.

---

### Review · Reviewer_hhVc · 2024-08-29

**Summary Of Contributions:**

This paper proposes an approach for Open Set Recognition/OOD detection. Concretely, a number of existing methods (from OOD detection/Contrastive Learning etc.) are combined synergistically resulting in improved detection of unknown classes. The basic idea is to increase intra-class compactness and inter-class sparsity in the hope that OOD samples will more likely fall into the larger volume between known class centres in the feature space leading to better detection.

**Audience:**

Yes

**Claims And Evidence:**

Yes

**Requested Changes:**

1. (Critical) Add proper comparison to CIDER [1], both in terms of discussing the similarities and differences to S2CA, as well empirical results.
2. (Critical) Add discussion with regards to real-world object detection and how S2CA may be applied. (Non-critical) Compare with [2] on their 2-stage object detector benchmark (which is fully open source)
3. (Critical) Add discussion on Failure Detection including references to [3,4], e.g. as related/future work. (Non-critical) Evaluate S2CA on failure detection.
4. (Non-critical) Make clear in the tables which approaches require altering the training procedure. Also ablated difference choices of post-training OOD score for S2CA.

**Strengths And Weaknesses:**

Apologies as I am quite pressed for time, so I may have misunderstood/missed information in the paper. I welcome the authors to correct me if that is the case.

**Strengths**
- The approach is relatively simple whilst being seemingly effective. The authors present a clear ablation of their approach, demonstrating how each component contributes to the overall performance.
- The paper benchmarks lidar data, which is typically overlooked in the field of OSR/OOD detection.

**Weaknesses**
1. The core idea of increasing intra-class compactness and inter-class sparsity has been proposed before in [1], which is not referenced at all in the paper.
2. The choice to extract objects using ground truth bounding boxes is a bit odd. This effectively reduces the problem of detection down to classification. The paper does not perform any actual object detection experiments on 2-stage object detectors, for which the proposed method should be readily applicable [2].
3. Missing discussion on Failure Detection [3,4]. Recent work has proposed an alternative perspective where other misclassified samples of known classes and samples of unknown samples should be detected. This is not mentioned at all in the paper, even though, in scenarios such as autonomous driving, the misclassification of a known class could be just as problematic as the treatment of an unknown class as a know one.
4. It's unclear in Tab. 2 which approaches are training-based and which are post-training. It's also unclear whether choosing a different post-training score rather than ViM (e.g. Mahalanobis distance) would benefit S2CA.


[1] Ming et al. *How to Exploit Hyperspherical Embeddings for Out-of-Distribution Detection?* ICLR 2023

[2] Du et al. *VOS: Learning What You Don't Know by Virtual Outlier Synthesis* ICLR 2022

[3] Jaeger et al. *A Call to Reflect on Evaluation Practices for Failure Detection in Image Classification* ICLR 2023

[4] Xia et al. *Augmenting Softmax Information for Selective Classification with Out-of-Distribution Data* ACCV 2022

---

> ### Author Response · Authors · 2024-09-16
> **Response to Reviewer hhVc**
>
> Thank you for your comments. Please see our responses below:
>
> 1.	We have included a reference to CIDER and added a discussion in section 5.3 to explain the differences between CIDER and the proposed S2CA. An empirical comparison, however, requires training CIDER on our data, which is not feasible in two weeks. We also believe that this comparison will not produce any additional insight.
>
> 2.	We have added an explanation in the conclusion section on the application of the proposed method in real-world scenarios, where S2CA can be applied in the classification step in two-step object detectors. We feel an empirical evaluation of the method for open set object detection is out of scope, as our method focuses on the recognition task rather than detection.
>
> 3.	We have included a discussion on failure detection as a future work in the conclusion section.
>
> 4.	We have updated Table 2 to show which methods are trainable. We also provide an ablation study of different post-training OOD scores in Table 5. Our results confirm that ViM is the best performing OOD score as it combines class-agnostic information in the feature space and the class-dependent information in the logit space, corresponding to the S2CA loss which is applied to both feature space and logit space.

---

> > ### Comment · Reviewer_hhVc · 2024-09-19
> >
> > Thanks for taking the time to address my considerations.
> > I apologise for the delayed reply.
> >
> > > An empirical comparison, however, requires training CIDER on our data, which is not feasible in two weeks.
> >
> > I would be happy with a comparison using the experimental set up from CIDER https://github.com/deeplearning-wisc/cider on CIFAR, which should be fast to train (<1 day on a single GPU).
> >
> > > We also believe that this comparison will not produce any additional insight.
> > Both approaches aim to address/improve upon deficiencies found with supervised contrastive learning in the context of OSR, by using (different) anchor-based learning paradigms. I believe it is of interest to the community to see them compared and contrasted empirically.
> >
> > Unfortunately I am still inclined towards reject if you are unable to provide any empirical comparison between your work and CIDER.
> >
> > As I recognise I have replied late in the discussion period maybe the action editor could give the authors a little more time to add these results?

---

> > > ### Comment · Action_Editor_DwJY · 2024-09-19
> > >
> > > I am happy to provide the authors with three weeks to produce the necessary empirical comparisons to make the paper as strong as possible.
> > > Action Editor

---

> > > ### Author Response · Authors · 2024-09-23
> > > **Response to Reviewer hhVc**
> > >
> > > Thank you for your comments. We have included the empirical comparison between our method and CIDER in Section 5.3.

---

> > > > ### Comment · Reviewer_hhVc · 2024-09-23
> > > >
> > > > Thanks for providing the additional results. I believe the paper is much improved and I am happy with the paper now.

---

### Review · Reviewer_Aes7 · 2024-09-04

**Summary Of Contributions:**

In this paper, the Open Set Recognition problem is examined, to reject unknown objects. The paper tackles this problem by using a combination of supervised contrastive learning (class anchoring) and classifier learning (minimizing distance to correct class, maximizing distance to incorrect class) - together with a weighting between the two branches, together with the virtual logit methodology in [1]. The model is then run on autonomous driving datasets (nuScenes and KITTI) and performance is recorded for open-set recognition.

**Audience:**

Yes

**Claims And Evidence:**

Yes

**Requested Changes:**

On the lines indicated above, can the authors run more ablation analysis showing the effect of the various components used in the modelling?

**Strengths And Weaknesses:**

+ This is a principled way to construct the problem through a combination of contrastive, supervised and virtual logits.
+ The results are definitely convincing, outperforming existing methods. The evaluations test against different modelling backbones and show consistency in performance.
- Novelty appears to be slightly lacking, as it looks like a mix and match approach from different sources.
- I think the analysis could use more ablations. I am looking at the impact of the different pieces used in the modelling (e.g. supervised, contrastive).
- More insight would be useful as to how each contributes to the performance.

---

> ### Author Response · Authors · 2024-09-16
> **Response to Reviewer Aes7**
>
> Thank you for your comments. We have revised the introduction section to highlight the novelty of the paper.  We have also included additional ablations in section 5.2 to show the effectiveness of each module.

---

> > ### Comment · Reviewer_Aes7 · 2024-09-21
> > **Convincing ablations**
> >
> > These are very convincing results with the ablations. Thanks for adding.

---

### Decision · Action_Editor_DwJY · 2024-09-25

**Recommendation:** Accept as is

**Comment:**

The proposal of Supervised Contrastive Class Anchor overcomes some of the limitation of Open Set Recognition (OSR) methods. The experimental results on two widely used driving scene recognition datasets are convincing. The reviewers, while not extremely enthusiastic about the paper, recommend that it be accepted as some researchers in the community would benefit from the methods introduced herein.

**Audience:**

Yes. Some of TMLR's audience (empirical researchers interested in self-driving applications) would be interested in the findings of the paper.

**Claims And Evidence:**

The experimental results on two widely used driving scene recognition datasets (i.e., KITTI and nuScenes) are comprehensive and support the claims of the submission.